# Improvement Performance of p-GaN Gate High-Electron-Mobility Transistors with GaN/AlN/AlGaN Barrier Structure

**DOI:** 10.3390/mi15040517

**Published:** 2024-04-12

**Authors:** An-Chen Liu, Yu-Wen Huang, Hsin-Chu Chen, Hao-Chung Kuo

**Affiliations:** 1Department of Photonics and Institute of Electro-Optical Engineering, College of Electrical and Computer Engineering, National Chiao Tung University, Hsinchu 30010, Taiwan; 2Institute of Advanced Semiconductor Packaging and Testing, National Sun Yat-sen University, Kaohsiung 804201, Taiwan; 3Semiconductor Research Center, Hon Hai Research Institute, Taipei 114699, Taiwan

**Keywords:** p-GaN gate, HEMT, epitaxy, threshold voltage, gate leakage current

## Abstract

This study demonstrates a particular composited barrier structure of high-electron-mobility transistors (HEMTs) with an enhancement mode composed of p-GaN/GaN/AlN/AlGaN/GaN. The purpose of the composite barrier structure device is to increase the maximum drain current, reduce gate leakage, and achieve lower on-resistance (R_on_) performance. A comparison was made between the conventional device without the composited barrier and the device with the composited barrier structure. The maximum drain current is significantly increased by 37%, and R_on_ is significantly reduced by 23%, highlighting the synergistic impact of the composite barrier structure on device performance improvement. This reason can be attributed to the undoped GaN (u-GaN) barrier layer beneath p-GaN, which was introduced to mitigate Mg diffusion in the capping layer, thus addressing its negative effects. Furthermore, the AlN barrier layer exhibits enhanced electrical properties, which can be attributed to the critical role of high-energy-gap properties that increase the 2DEG carrier density and block leakage pathways. These traps impact the device behavior mechanism, and the simulation for a more in-depth analysis of how the composited barrier structure brings improvement is introduced using Synopsys Sentaurus TCAD.

## 1. Introduction

In recent years, GaN wide bandgap (WBG) semiconductors have emerged as frontrunners in the quest for advanced electronic materials, surpassing traditional silicon-based counterparts with higher breakdown voltages (BV) and R_ON_ [1]. The unique physical properties of GaN, especially the formation of a two-dimensional electron gas (2DEG) at the AlGaN/GaN interface due to spontaneous and piezoelectric polarizations, positions GaN HEMTs as intrinsic depletion-mode devices. These devices are characterized by a normally on behavior with a threshold voltage (V_TH_) of less than 0 V, complicating their integration into power circuits due to the need for an external bias to deactivate the channel [2,3]. This challenge led to research toward enhancement-mode (E-mode) GaN HEMTs, which operate under a positive threshold voltage, thereby simplifying their application in power converters and reducing circuit design complexity.

To address these challenges and control the full potential of GaN HEMTs, innovative approaches were explored to manipulate the electrical properties and structural characteristics of GaN-based devices. Among these strategies were the development of non-polar a-plane channels, fluorine treatment [4], gate recesses [5], p-type GaN cap [6,7,8,9,10], and cascode structures. Among these approaches, the p-GaN gate structure has become a commercially available normally off p-GaN gate HEMT due to its outstanding figure of merit and reliable normally off functionality for power switching applications that require normally off characteristics, such as CMOS circuits that require safe operation and simple gate drive configurations. Therefore, there are several significant challenges facing p-GaN gate HEMTs that require improvement. Firstly, reducing gate leakage current by adding a dielectric layer under the gate is an effective strategy to reduce leakage current and increase gate drive voltage [11]. Secondly, during the growth of the p-GaN cap layer and to ensure optimal activation via MOCVD, it is necessary to incorporate an intrinsic GaN layer. This layer functions as a barrier to minimize Mg out-diffusion into the AlGaN barrier and/or GaN channel, thus preventing 2DEG degradation [12].

Additionally, this study delved into the critical role of epitaxial layer engineering in optimizing device performance [11,12]. The introduction of epitaxial considerations, such as the use of AlN spacer layers [13,14] and the intrinsic u-GaN barrier layer, aimed to enhance 2DEG density and electron mobility while mitigating the adverse effects of Mg diffusion from p-type GaN cap layers [15,16,17,18]. This epitaxial layer engineering not only improves device efficiency but also solves critical reliability issues, representing a key strategy for GaN HEMTs in high-power and high-frequency applications. Via this comprehensive approach, the study contributed to the ongoing advancements in GaN technology, paving the way for more efficient, reliable, and simplified power systems.

In this study, a p-type cap layer technique was implemented to convert depletion-mode HEMTs into E-mode HEMTs, achieving outstanding performance, reliability, and commercial potential. The introduction of a p-GaN gate layer modified the energy levels in the AlGaN, effectively depleting the 2DEG channel at zero gate bias. This operation relied on careful gate stack design, influencing the positive V_TH_ and the gate-source-drain current (I_GSS_). For effective 2DEG depletion at zero gate voltage (V_G_), the thickness of the AlGaN layer was optimized to 10–15 nm, with the p-GaN gate layer being 50–100 nm thick and a Mg doping concentration ranging from 10^18^ to 10^19^ cm^−^^3^ [19]. The epitaxial growth of the p-GaN layer, conducted at high temperatures around 1000 °C [20], introduced Mg ions as the p-type dopant. However, this process presented challenges such as Mg diffusion into the AlGaN and GaN channels, leading to increased P-i-N diode leakage and degradation of the 2DEG [21], thereby impacting device performance. The migration of Mg from the p-GaN into adjacent layers contributed to the observed trapping effects and aggravated P-i-N diode leakage [21], distinguishing the need for optimized doping and layer design strategies to mitigate these issues. The objective was to address Mg outward diffusion by reducing Mg doping concentrations, lowering activation temperatures, or even utilizing epitaxy techniques. However, the first and second solutions resulted in a lower threshold voltage, causing the HEMT to be inoperable in the E-mode configuration. The use of an AlN spacer layer was strategic for enhancing the interface polarization effects, leading to an increased generation of polarized carriers. This enhancement was primarily due to the AlN high energy gap, which served to effectively block leakage pathways by minimizing electron tunneling across the interface. The integration of AlN spacer layers above and below the crucial epitaxial structures served distinct purposes: the layer above aimed to improve the confinement of the 2DEG by enhancing polarization effects, thus increasing carrier density. In contrast, the layer below acted as a barrier to prevent the diffusion of dopants or defects from the substrate, further contributing to the device’s overall performance and stability.

The lack of an AlN spacer layer in the device structure brings some disadvantages. Without the enhanced interfacial polarization provided by these spacer layers, the generation of polarized carriers is reduced, which significantly limits 2DEG density. This limitation directly affects the conductivity and performance of the device. Furthermore, without the AlN spacer, there is a lack of barrier to the leakage path, resulting in higher leakage current, which in turn affects device efficiency. Additionally, without the protective barrier provided by the bottom AlN layer, there was an increased risk of dopant or defect diffusion from the substrate, which could further degrade device performance over time. In the study, the AlGaN/GaN HEMTs featured a p-GaN gate, complemented by the introduction of a u-GaN barrier layer, undoped, strategically placed between the p-GaN cap and the AlGaN barrier. This configuration was further enhanced by the inclusion of an AlN spacer layer. The u-GaN barrier layer acted as an effective diffusion barrier, crucially mitigating the negative impacts of Mg diffusion from the p-GaN cap into the AlGaN barrier, thereby significantly reducing the P-i-N leakage current. This method addressed critical issues related to Mg migration, enhancing device reliability and performance.

The 2DEG configuration was subjected to assorted scattering phenomena, each contributing differently to the 2DEG mobility. The array of scattering origins encompasses background contaminants, acoustic phonon interference, distortion diffusion due to stress fields around the dislocations deformation potential, interface irregularities at the AlGaN/GaN heterojunction, and compound dispersion resulting from the restricted penetration of the electron wavefunction into the AlGaN barrier. Upon integrating an AlN layer, which ranged from 0.7 to 2 nm in thickness, into the AlGaN/GaN HEMT framework, the assembly was transformed into an AlGaN/AlN/GaN HEMT junction. The primary intention behind this insertion was to enhance the energy band discontinuity between AlGaN and GaN, thereby obstructing the incursion of channel electrons into AlGaN. Such a modification elevated the quantum well-confining capabilities and augmented the electron density.

Moreover, the addition of the AlN spacer layer played a pivotal role in increasing the 2DEG carrier density. The high energy gap of the AlN spacer not only blocked leakage pathways but also enhanced interface polarization effects. This resulted in a substantial increase in polarized carriers, significantly improving electron mobility within the device. The combined effects of the u-GaN barrier layer and AlN spacer layer not only solved the problem of Mg diffusion but also led to improved electron mobility by increasing 2DEG carrier density. These advancements collectively contributed to the development of more efficient, reliable, and high-performing AlGaN/GaN HEMTs.

The study of p-GaN gate breakdown behavior under forward bias conditions has attracted widespread attention in this field. Special attention was dedicated to understanding this breakdown phenomenon in [22], where breakdown events were observed to be associated with the formation of percolation paths within the depletion region of the p-GaN layer, especially in the region close to the metal interface. Similarly, [23] provided insights into the breakdown mechanism by emphasizing avalanche multiplication within the space charge region of the Schottky metal/p-GaN junction. Furthermore, experimental observations combined with simulation studies presented in [24,25] reveal the impact of high electric fields within the p-GaN layer on p-GaN HEMT gate reliability. This study highlights the critical role of understanding breakdown mechanisms and associated degradation phenomena in Schottky metal/p-GaN device performance and reliability.

In this article, we investigate an alternative approach to improve p-GaN HEMT performance by introducing the u-GaN barrier layer between the p-GaN and AlGaN/GaN layers. The main reason for introducing the u-GaN barrier layer was the need to suppress Mg diffusion into AlGaN/GaN. This approach is designed to increase the maximum drain current, reduce gate leakage, and achieve lower on-resistance (R_on_) characteristics. Without the u-GaN barrier layer, Mg diffusion was more pronounced, leading to undesirable changes in the threshold voltage and carrier concentration. Thus, the u-GaN layer served as a critical barrier, ensuring the maintenance of the desired electronic characteristics and improving the overall device reliability.

## 2. Materials and Methods

The fabrication of p-GaN HEMTs adhered to a standardized base structure, incorporating an identical AlN seed layer, a GaN/AlGaN buffer layer, and an Al_0.18_Ga_0.82_N barrier layer across all device variants. Device performance variations were introduced via modifications to the epitaxial structures of the active layers, designed to explore the impact of the AlN spacer and u-GaN barrier layer on device characteristics. The epitaxial design of the active layers was categorized into three distinct structures for comparative analysis: Structure (a) comprised devices without an AlN spacer and u-GaN layer; Structure (b) included devices featuring an AlN spacer but without a u-GaN layer; Structure (c) consisted of devices incorporating both an AlN spacer and a u-GaN layer. These variations were elaborately described in Table 1, which detailed the different p-GaN HEMTs structures and doping conditions, offering a comprehensive overview of the design parameters for each configuration. Furthermore, Figure 1 graphically depicts these structural distinctions, illustrating the comparative layout of the epitaxial layers and the specific modifications applied to each device structure. This design and fabrication methodology facilitated a systematic investigation into the effects of AlN spacer and u-GaN layer integration on the electrical performance and efficiency of p-GaN HEMTs.

The u-GaN barrier primarily function serves to suppress Mg diffusion from p-GaN to the internal structure of AlGaN/GaN, and the u-GaN barrier was fabricated using Metalorganic Chemical Vapor Deposition (MOCVD) employing highly uniform epitaxial technology. It exhibits high crystal quality and low defect density characteristics. Therefore, by introducing the u-GaN barrier under the p-GaN gate, the diffusion of Mg during the high-temperature process can be effectively controlled, reducing the formation of additional p-type hole defects and further improving gate leakage.

Additionally, we found that some researchers focused on improving surface states, particularly via improvements in recessed gate AlGaN/GaN MISHEMTs. For the case of gate leakage current caused by surface states, low-damage etching techniques were required, such as atomic layer etching (ALE) [26] or neutral beam etching (NBE) [27]. These etching techniques also provide precise control of the recessed gate depth due to the atomic-scale etching reaction mechanism. However, the challenge of precisely controlling atomic-level reactive etching uniformly across large areas, especially in critical gate regions, remains an important issue at present. This is crucial as the varying depths of recessed gates can significantly impact electrical performance, such as V_TH_ shifting and high–low drain current. After recessed gate etching, ensuring the quality of the gate dielectric becomes a critical issue. This is because the growth of the gate dielectric introduces donor-like defects in the AlGaN surface. Donor-like defects will cause gate leakage current and subsequent current collapse. Therefore, Liu et al. demonstrated that passivating the interface of AlGaN before growing the gate dielectric is crucial in mitigating these issues [28].

The fabrication of E-mode p-GaN HEMTs utilizing a gate-first methodology commenced by crafting a p-GaN mesa via inductively coupled plasma–reactive ion etching (ICP-RIE), followed by mesa isolation achieved via ion implantation to selectively etch designated areas. Subsequently, the formation of the Schottky contact (gate metal) was accomplished using titanium/aluminum/nickel (Ti/Al/Ni) layers with thicknesses of 25/50/5 nm, respectively, deposited by e-gun sputtering. Following the deposition of gate metal, a sequence of surface preparation involving a wet cleaning process with hydrochloric acid (HCl)/deionized water (DI) at a 1:10 ratio, and remote plasma pretreatment was employed to eliminate native oxides. This preparation was followed by the atomic layer deposition (ALD) of a 15 nm thick high-dielectric Al_2_O_3_ layer, synthesized from trimethylaluminum and H_2_O precursors at a temperature of 350 °C. Additionally, a 50 nm Si_3_N_4_ passivation layer was deposited using plasma-enhanced chemical vapor deposition (PECVD).

The fabrication proceeded with the establishment of ohmic contacts (source/drain metal) using a Ti/Al/Ti stack of 25/125/45 nm thicknesses, solidified by rapid thermal annealing (RTA) at 825 °C for 30 s under a nitrogen (N_2_) atmosphere to finalize the source-drain ohmic connections. The resistance of these ohmic contacts was measured at 0.6 Ω-mm. Subsequent steps follow the back-end process, including the gate field plate, 150 nm Si_3_N_4_ passivation layer, and contact openings. The final process was to deposit a Ti/Al (25 nm/300 nm) metal stack as the pad electrodes. The featured size of the p-GaN device was as follows: gate length (L_G_) is 3 μm, gate-to-source distance (L_GS_) is 3 μm, gate-to-drain distance (L_GD_) is 10 μm, and gate width (W_G_) is 100 μm, as shown in Figure 1e.

## 3. Results and Discussion

In the DC measurements of p-type GaN, HEMTs across three different structures were analyzed, revealing significant variations in gate control, drain current (I_D_), gate-source voltage (V_G_), and drain–source voltage (V_D_) characteristics, as illustrated in Figure 2. These devices exhibited substantial I_D_ currents and on/off ratios, reaching up to eight orders of magnitude, highlighting the high-quality epitaxial conditions and stability of the p-GaN HEMTs fabricated. The study aimed to analyze the transfer and gate leakage characteristics of p-GaN E-mode HEMTs with varying epitaxial structures, employing a device design with dimensions L_G_/L_GS_/L_GD_/W_G_ of 3/3/10/100 μm for the drain current–gate bias (I_D_-V_G_) measurement. The results, depicted in Figure 2, demonstrated linear scale transmission characteristics. V_TH_ for devices A, B, and C were determined to be 1.5 V, 1.4 V, and 1.0 V, respectively, using a linear extrapolation method based on the maximum transconductance curve gate bias. Furthermore, the maximum drain current (I_D, max_) at a V_G_ of 6 V was observed to be 198 mA/mm for Device A, 241 mA/mm for Device B, and 272 mA/mm for Device C, with maximum transconductance G_m, max_ values of 51.2 mS/mm, 72.7 mS/mm, and 77.3 mS/mm, respectively.

The semi-logarithmic I_D_-V_G_ curves, with a V_D_ operating at 10 V and V_G_ sweeping from −4 V to 6 V, are depicted in Figure 2a. Subthreshold slope (SS) values of 121 mV/dec for Device A, 117 mV/dec for Device B, and 102 mV/dec for Device C were observed. Among Device C was superior gate control efficiency, attributed to the reduced Mg diffusion into the barrier and decreased band-to-band leakage via the introduction of AlN spacers.

Figure 2b presents the conduction resistance of the devices, with on-resistance values for Devices A, B, and C of 14.5 Ω-mm, 12.1 Ω-mm, and 11.1 Ω-mm, respectively. Devices B and C, incorporating AlN spacer layers, exhibited lower conduction resistance compared to Device A. Notably, Device C, featuring an effective AlN spacer layer and a thicker u-GaN barrier layer, prevented Mg diffusion into AlGaN. This configuration reduced defects and minimized leakage in the device, substantially reducing conduction resistance and improving device performance. The introduction of AlN spacer layers was aimed at enhancing interface polarization fields, injecting more electrons into the 2DEG channel, thereby reducing channel impedance and improving turn-on resistance. The u-GaN barrier layer design can provide multiple functions: it alleviates the weakening of the p-GaN: Mg hole on the gate control capability V_TH_ and acts as a barrier to slow down Mg diffusion, solving the core challenge of p-GaN HEMT devices. This study emphasizes the critical value of solving magnesium diffusion issues and utilizing AlN spacers to improve device efficiency.

In Figure 2b, the current collapse was observed under high V_GS_ and V_DS_ conditions, which could be attributed to the high electric field effect between the gate and drain. This effect is exacerbated by the larger lattice mismatch at the GaN-on-Si HEMT heterojunction interface, resulting in a higher density of threading dislocations within the heteroepitaxial layer. To solve the current collapse problem, several effective strategies will be considered and improved in further study. These include optimized u-GaN barrier thickness, source-to-drain field plate (FP) design, plasma treatment, and surface passivation [29]. In addition, the self-heating effect under high-power operation may cause current collapse, especially at high voltages, and thermal management becomes a key factor limiting device performance [30].

In the configurations of Devices B and C, the inclusion of an AlN spacer layer resulted in a reduction in on-resistance compared to Device A, as demonstrated in the analysis. Specifically, Device C incorporated an AlN spacer layer adept at efficiently blocking leakage pathways, coupled with the u-GaN barrier layer that enhanced the barrier thickness. This enhancement reduces defects due to Mg migration into AlGaN and mitigates leakage in P-i-N diodes due to Mg outflow. As a result, the on-resistance was significantly reduced, and I_G_ was improved, as shown in Table 2. This design highlights the dual benefits of integrating AlN spacers and u-GaN barrier layers: enhanced device integrity by reducing leakage and defects and improved electrical performance by reducing on-resistance.

Synopsys Sentaurus TCAD simulations were introduced to study different epitaxial structures, both with and without AlN spacers and the u-GaN barrier layer, for the analysis of electronic properties in p-GaN HEMTs. Figure 3 displays the key layers of the three distinct structures under simulated conditions. The simulations incorporated device dimension parameters, including L_G_ of 3 μm, L_GS_ of 3 μm, and L_GD_ of 10 μm. With the increase in V_G_ above zero, the depletion region width was observed to decrease rapidly, thereby enhancing the 2DEG concentration in the channel. The simulated Mg doping concentration in p-GaN was set to 9 × 10^18^ cm^−3^, closely simulating the actual device characteristics. Ohmic contacts were used for the source and drain, while the gate employed Schottky contact, determined by the work function difference between the metal and the semiconductor. This simulation work aims to emphasize the optimization fraction of the study, incorporating literature-based physical theory to enhance the understanding of the physical mechanisms involved. The introduction of AlN spacers utilizes the polarization effect to increase the 2DEG current density, thus enhancing electron mobility. Furthermore, the integration of the u-GaN barrier layer served to suppress Mg diffusion, reducing defects and consequently improving the SS. This comprehensive analysis highlights the key points of epitaxial structural optimization to improve the performance and reliability of p-GaN HEMTs by improving electronic properties and reducing leakage and defects.

Figure 3 demonstrates that, with the inclusion of AlGaN/GaN interface traps in the analysis, the distance between the Fermi level and the bottoms of the three conduction bands widened. This phenomenon was observed across Devices A, B, and C, with all devices showing significant increases in this distance despite Device C exhibiting the smallest V_TH_. This observation indicated that the variations in V_TH_ among the devices were primarily attributed to the presence of AlGaN/GaN interface traps rather than differences in the band structure. The creation of interface traps at the AlGaN/GaN interface can be explained by the discontinuities and irregularities at the material boundary, which disrupt the crystal structure and create localized energy states within the bandgap. These states can trap carriers, influencing the device’s electrical properties by affecting the Fermi level position and, consequently, the threshold voltage of the device. Interface traps introduce a complex mechanism that impacts device performance, including threshold voltage shifts, decreased mobility due to scattering, and increased leakage currents.

The analysis and simulation of the density of interface states (*Dit*) mechanism provided a clearer understanding of how these traps influence device behavior. By calculating the *Dit*, the study aimed to quantify the impact of interface traps on the electronic properties of the HEMTs, providing insights into how they contribute to performance variability and identifying strategies for mitigating their effects. This approach underscores the importance of interface engineering in optimizing semiconductor device performance and reliability.

Considering a continuum of trap levels, the conductance parameter $G_{p}$ can be expressed as follows:(1)$$\frac{G_{p}}{\omega }=\frac{qDit}{2\omega \tau_{it}}ln\left[1+{\left(\omega \tau_{it}\right)}^{2}\right]$$

In this equation, ω represents the radial frequency, and $\tau_{it}$ denotes the trap time constant given by the Shockley–Read–Hall statistics [31].
(2)$$Dit\approx \frac{2.5}{Aq}{\left[\frac{G_{p}}{\omega }\right]}_{max}$$

An approximate expression gives the *Dit* interface trap densities in terms of the measured maximum conductance, with A representing device area, and q denoting the elementary charge. According to the conductance measurement data, *Dit* and its corresponding trap energy level located below the conduction band are obtained by fitting the data with Equations (1) and (2) [32].

Figure 4 depicts an elaborate evaluation of the density of interface states in relation to the energy discrepancy between the conduction band edge (EC) and the trap energy level (ET), revealing that Device C exhibited a substantial decline in the density of *Dit* adjacent to the conduction band within shallow energy states, in stark contrast to Devices A and B. This notable decrease is indicative of a significant enhancement in the quality of the interface, a factor that is essential for the operational efficacy of the device. An assessment of *Dit* against the energy variation (∆E) was methodically carried out for all devices utilizing the conductance method by Equations (1) and (2). The results disclosed interface trap densities for Devices A, B, and C as 1.75 × 10^14^ to 2.85 × 10^13^ cm^−2^ eV^−1^, 6.44 × 10^13^ to 2.09 × 10^13^ cm^−2^ eV^−1^, and 3.6 × 10^13^ to 2.21 × 10^13^ cm^−2^ eV^−1^, respectively, as illustrated in Figure 4. These findings highlight the disparities in interface quality among the devices, with Device C achieving the most optimal result in terms of diminished trap density.

Mg on Ga substitutional acceptor complexes are known to align their energy levels near the conduction band, ranging between 0.26 eV and 0.6 eV. This closeness to the conduction band plays a crucial role as it directly impacts the electrical properties and the efficiency of charge carrier movement within the device. The comparative study demonstrated a pronounced reduction in defect density at the AlGaN/GaN interface for Devices B and C when compared to Device A, as illustrated in Figure 4. Such findings corroborate the theory that the incorporation of an u-GaN barrier layer and an AlN spacer layer is an efficacious approach to overcoming the challenges associated with Mg diffusion in enhancement-mode devices that include a P-GaN cap layer. This deliberate structural adaptation not only addresses issues related to diffusion-induced defects but also substantially enhances the device’s performance and the semiconductor interface’s quality, thereby establishing a direct link between the architectural modifications and the noted advancements in device functionality.

This study confirmed that the inclusion of AlGaN/GaN interface traps resulted in V_TH_ for all three structures being higher than those observed in the absence of such traps, indicating that interface traps were the cause of the V_TH_ increase. In reality, the presence of traps and the introduction of an u-GaN barrier layer, which thickens the barrier, were found to suppress traps within the AlGaN layer and reduce P-i-N leakage. This modification brought the V_TH_ closer to values seen without AlGaN/GaN interface traps. Although the improvement in leakage current was modest, the variations in V_TH_ were significant. The design adjustments led to a substantial increase in current magnitude by a certain percentage and an improvement in the S.S. by a specific percentage while maintaining V_TH_ above 1 V, ensuring that the design outcomes remained within acceptable standards. This approach highlighted the effectiveness of strategic layer integration and trap management in optimizing device performance, particularly in terms of enhancing electrical characteristics while adhering to operational thresholds.

## 4. Conclusions

In this study, we successfully demonstrated the composited barrier structure of GaN/AlN/AlGaN to improve E-mode p-GaN HEMT performance. Compared with the conventional p-GaN/AlN/AlGaN barrier structure with Device A, the composite barrier structure with Device C can effectively increase the maximum drain current by 37% and reduce R_on_ by 23%. Notably, Device C surpassed Device B in performance, as the u-GaN barrier layer effectively blocked Mg diffusion into the AlGaN barrier layer, leading to a decrease in P-i-N leakage and gate leakage improvement. The decrease in V_TH_ indicates an increase in the ability of gate control, mainly due to the AlN barrier leading to more polarization effects, producing more 2DEG concentration in the channel. These results are consistent with TCAD simulations. The study confirmed that the presence of traps and the introduction of the u-GaN barrier layer, which thickens, is found to suppress traps within the AlGaN layer and reduce P-i-N leakage. Finally, we believe that the potential composite barrier structure of p-GaN/GaN/AlN/AlGaN for GaN HEMTs is promising for third-generation semiconductor applications. It points in a promising direction for the future development of semiconductor technology.

## Figures and Tables

**Figure 1 micromachines-15-00517-f001:**
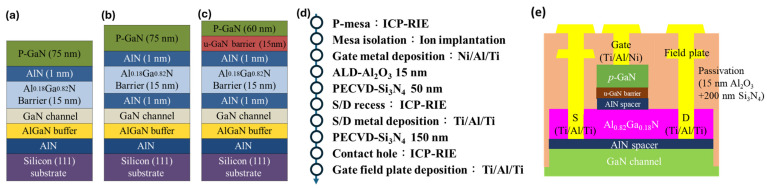
Epitaxial structure of p-GaN HEMTs: (**a**) normally epitaxial structure, (**b**) with AlN spacer, and (**c**) with AlN spacer & u-GaN layer. (**d**) Fabrication flow of p-GaN gate devices. (**e**) Cross-section schematic view of the p-GaN HEMT (Device C).

**Figure 2 micromachines-15-00517-f002:**
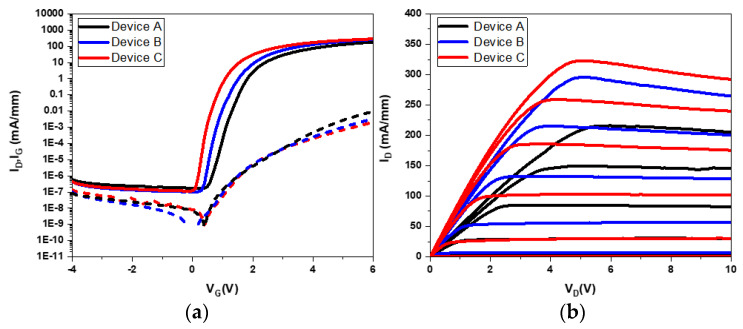
The DC measurement curve of (**a**) I_D_, I_G_-V_G_ (V_D_ = 10 V), and (**b**) I_D_-V_D_ (V_G_ = −4~6 V, step = 2 V).

**Figure 3 micromachines-15-00517-f003:**
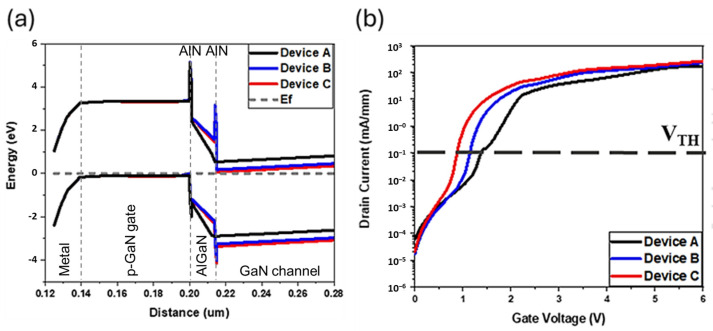
Setting of the AlGaN/GaN interface trap situation: (**a**) Energy band diagram; (**b**) I_D_-V_G_ characteristic curve (simulation).

**Figure 4 micromachines-15-00517-f004:**
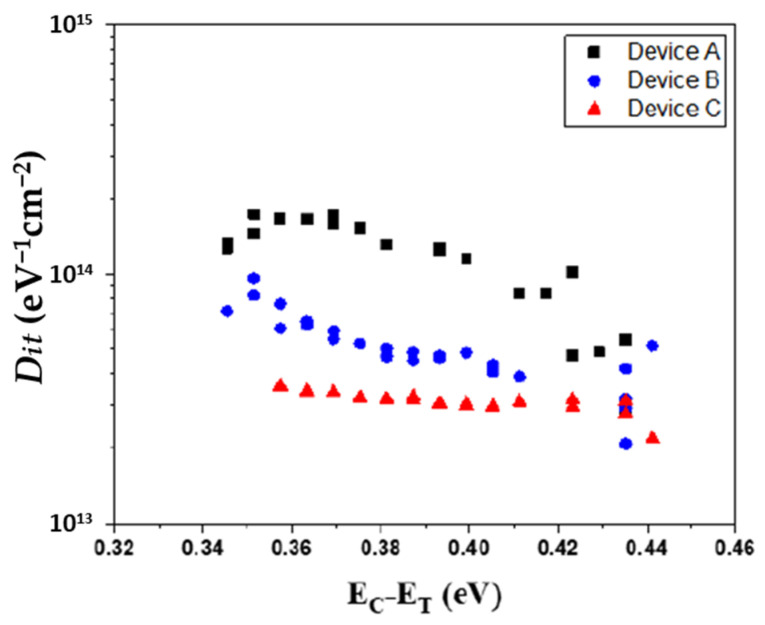
Depicts the derived density of interface traps against the energy difference between the E_C_-E_T_ for Device A, Device B, and Device C.

**Table 1 micromachines-15-00517-t001:** p-GaN/AlGaN/GaN heterostructures under different epitaxial conditions.

Devices/Conditions	A	B	C
p-GaN temperature (°C)	1005	1005	1005
p-GaN underlying u-GaN with or without	without	without	with
Mg growth flow (sccm)	225	225	225
Average Mg concentration (cm^−3^)	9 × 10^18^	9 × 10^18^	9 × 10^18^
Al_X_Ga_1-X_N, x = ?	0.18	0.18	0.18
with or without 1 nm of AlN spacer layer	without	with	with

**Table 2 micromachines-15-00517-t002:** Different epitaxial conditions of direct current numerical values.

Device	I_D,off_@V_G_ = −4 V (mA/m)	I_D,on_@V_G_ = 6 V (mA/mm)	I_G, on_ @V_G_ = 6 V @V_D_ = 10 V (mA/mm)	V_TH_@I_D_ = 0.1 mA (V)	SS (mV/dec)	G_m,max_ (mS/mm)
A	6.2 × 10^−7^	198	6.2 × 10^−3^	1.5	121	51.2
B	4.8 × 10^−7^	241	4.8 × 10^−3^	1.4	117	72.7
C	4.9 × 10^−7^	272	4.7 × 10^−3^	1	102	77.3

## Data Availability

The original contributions presented in the study are included in the article, further inquiries can be directed to the corresponding authors.

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
