# Peer review of "Improvement Performance of p-GaN Gate High-Electron-Mobility Transistors with GaN/AlN/AlGaN Barrier Structure"

_micromachines, 2024, doi:10.3390/mi15040517_

Round 1
Reviewer 1 Report
Comments and Suggestions for Authors
1. In Fig. 2, all the controlled devices show current collapse at high Vds. Please explain why.
2. All of the figures are not clear in the manuscript.
3. U-GaN is used to lower gate leakage current, which is the disadvantage of p-GaN HEMT. How about compared with the recessed gate AlGaN/GaN HEMT (Appl. Phys. Lett. 2021, 118, 173504; IEEE TRANSACTIONS ON ELECTRON DEVICES, VOL. 70, NO. 6, 2023), in which the gate leakage current can be suppressed by introducing a dielectric layer. The authors should make a comparison or at least make a comment on the above reported structures.
Author Response
- Comment: In Fig. 2, all the controlled devices show current collapse at high Vds, please explain why.
Reply: Thank you sincerely for your valuable suggestions. (P239-248)
In Figure 2(b), the current collapse was observed under high VGS and VDS conditions, which could be attributed to the high electric field effect between the gate and drain. This effect is exacerbated by the larger lattice mismatch at the GaN-on-Si HEMT heterojunction interface, resulting in a higher density of threading dislocations within the heteroepitaxial layer. To solve the current collapse problem, several effective strategies will be considered and improved in further study. These include optimized u-GaN barrier thickness, source-to-drain field plate (FP) design, plasma treatment, and surface passivation [28]. In addition, the self-heating effect under high-power operation may cause current collapse, especially at high voltages, and thermal management becomes a key factor limiting device performance [29].
[28] C. Deng et al., "Current collapse suppression in AlGaN/GaN HEMTs using dual-layer SiNx stressor passivation," Applied Physics Letters, vol. 122, no. 23, 2023.
[29] Q. Hu et al., "Improved Current Collapse in Recessed AlGaN/GaN MOS-HEMTs by Interface and Structure Engineering," IEEE Transactions on Electron Devices, vol. 66, no. 11, pp. 4591-4596, 2019.
- Comment: All of the figures are not clear in the manuscript.
Reply: Thank you for your suggestion. In response to the revision, we have provided further high-resolution Figures 1–4 in the manuscript.
- Comment: u-GaN is used to lower gate leakage current, which is the disadvantage of p-GaN HEMT. How about compared with the recessed gate AlGaN/GaN HEMT (Appl. Phys. Lett. 2021, 118, 173504; IEEE TRANSACTIONS ON ELECTRON DEVICES, VOL. 70, NO. 6, 2023), in which the gate leakage current can be suppressed by introducing a dielectric layer. The authors should make a comparison or at least make a comment on the above reported structures.
Reply: Thank you for your suggestion. To avoid confusion, we have redefined the u-GaN layer names as shown in Figure 1. Below the P-GaN is the u-GaN barrier, and below the AlGaN barrier is the GaN channel. (P151-171)
The u-GaN barrier primarily function serves to suppress Mg diffusion from p-GaN to the internal structure of AlGaN/GaN, and the u-GaN barrier was fabricated using Metalorganic Chemical Vapor Deposition (MOCVD) employing highly uniform epitaxial technology. It exhibits high crystal quality and low defect density characteristics. Therefore, by introducing the u-GaN barrier under the p-GaN gate, the diffusion of Mg during the high-temperature process can be effectively controlled, reducing the formation of additional p-type hole defects, and further improving gate leakage.
Additionally, we found that some researchers focused on improving surface states, particularly through improvements in recessed gate AlGaN/GaN MISHEMTs. For the case of gate leakage current caused by surface states, low-damage etching techniques were required, such as atomic layer etching (ALE) [25] or neutral beam etching (NBE)[26]. These etching techniques also provide precise control of the recessed gate depth due to the atomic-scale etching reaction mechanism. However, the challenge of precisely controlling atomic-level reactive etching uniformly across large areas, especially in critical gate regions, remains an important issue at present. This is crucial as the varying depths of recessed gates can significantly impact electrical performance, such as VTH shifting and high-low drain current. After recessed gate etching, ensuring the quality of the gate dielectric becomes a critical issue. This is because the growth of the gate dielectric introduces donor-like defects in the AlGaN surface. Donor-like defects will cause gate leakage current and subsequent current collapse. Therefore, Liu et al. demonstrated that passivating the interface of AlGaN before growing the gate dielectric is crucial in mitigating these issues [27].

Reviewer 2 Report
Comments and Suggestions for Authors
The introduction is excessively long. While it can be of general interest, it should focus more on positioning the article and justifying its relevance, rather than resembling a review manuscript. The authors must concentrate on highlighting the novelty of their work. Please adjust the Introduction accordingly.
As for Section 2, the authors should emphasize the differences between their approach and other comparable p-GaN technologies, e.g., the one by STM M. Moschetti et al., "Study of behavior of p-gate in power GaN under positive voltage", Proc. AEIT Int. Conf. Elect. Electron. Technol. Automot., pp. 1-6, 2020.
The authors just evaluate the impact on trapping, specifically the Ron under static excitation. Nevertheless, the authors are likely aware that trapping exhibits significant dynamics, which is crucial considering the typical operating regime of power circuits. Experimental results under dynamic operating conditions are typically conducted, as in Alemanno, A. et al. in "A Reconfigurable Setup for the On-Wafer Characterization of the Dynamic RON of 600 V GaN Switches at Variable Operating Regimes," Electronics 2023, 12, 1063. Have the authors conducted such tests? Could they provide commentary on this matter?
Author Response
Reviewer 2
- Comment: The introduction is excessively long. While it can be of general interest, it should focus more on positioning the article and justifying its relevance, rather than resembling a review manuscript. The authors must concentrate on highlighting the novelty of their work. Please adjust the Introduction accordingly.
Reply: Thank you for your suggestion. We have adjusted the introduction to focus on the novelty of this work, as shown below. (P125-133)
The study of p-GaN gate breakdown behavior under forward bias conditions has attracted widespread attention in this field. Special attention was dedicated to understanding this breakdown phenomenon in [Ref: Low frequency noise and gate bias instability in normally OFF AlGaN/GaN HEMTs], where breakdown events were observed to be associated with the formation of percolation paths within the depletion region of the p-GaN layer, especially in the region close to the metal interface. Similarly, [Ref: Investigation of the p-GaN Gate Breakdown in Forward-Biased GaN-Based Power HEMTs] provided insights into the breakdown mechanism by emphasizing avalanche multiplication within the space charge region of the Schottky metal/p-GaN junction. Furthermore, experimental observations combined with simulation studies presented in [Ref: Forward bias gate breakdown mechanism in enhancement-mode p-GaN gate AlGaN/GaN high-electron-mobility transistors, Ref: Study of the behavior of p-gate in Power GaN under positive voltage] reveal the impact of high electric fields within the p-GaN layer on p-GaN HEMT gate reliability. This study highlights the critical role of understanding breakdown mechanisms and associated degradation phenomena in Schottky metal/p-GaN device performance and reliability.
- Comment: As for Section 2, the authors should emphasize the differences between their approach and other comparable p-GaN technologies, e.g., the one by STM M. Moschetti et al., "Study of behavior of p-gate in power GaN under positive voltage", Proc. AEIT Int. Conf. Elect. Electron. Technol. Automot., pp. 1-6, 2020.
Reply: Thank you for your suggestion. We have added differences between this approach and other comparable p-GaN technologies as follows. (P114-124)
The study of p-GaN gate breakdown behavior under forward bias conditions has attracted widespread attention in this field. Special attention was dedicated to understanding this breakdown phenomenon in [21], where breakdown events were observed to be associated with the formation of percolation paths within the depletion region of the p-GaN layer, especially in the region close to the metal interface. Similarly, [22] provided insights into the breakdown mechanism by emphasizing avalanche multiplication within the space charge region of the Schottky metal/p-GaN junction. Furthermore, experimental observations combined with simulation studies presented in [23-24] reveal the impact of high electric fields within the p-GaN layer on p-GaN HEMT gate reliability. This study highlights the critical role of understanding breakdown mechanisms and associated degradation phenomena in Schottky metal/p-GaN device performance and reliability.
- Comment: The authors just evaluate the impact on trapping, specifically the Ron under static excitation. Nevertheless, the authors are likely aware that trapping exhibits significant dynamics, which is crucial considering the typical operating regime of power circuits. Experimental results under dynamic operating conditions are typically conducted, as in Alemanno, A. et al. in "A Reconfigurable Setup for the On-Wafer Characterization of the Dynamic RON of 600 V GaN Switches at Variable Operating Regimes," Electronics 2023, 12, 1063. Have the authors conducted such tests? Could they provide commentary on this matter?
(a) (b)
Reply: We thank the Reviewer for pointing us to references.
Figure (a) illustrates the dynamic degradation of Ron as measured in device C. Under these conditions, a blocking voltage of 200 V (VD,Block) is applied for 1 minute, while a high electric field was induced in the access area by applying the VG,OFF of -5 V, and then a switched bias state applied (VG,ON of 6V and VD,ON of 0-10 V). This combination triggers charge-trapping phenomena, allowing their impact on trapping to be evaluated by the dynamic variation of Ron. In Figure (b), this effect is already detectable at a VD,Block of 200V. It is observed that a slight increase in voltage corresponds to RD,ON reaching a degradation factor of 1.13. This indicates that typical transistor characteristics are retained. The observed degradation is significantly lower compared to an on-wafer 600V GaN switch where RD,ON reaches a degradation factor of approximately 2.3 at VD,Block of 200V [Ref Alemanno, A. et al. in "A Reconfigurable Setup for the On-Wafer Characterization of the Dynamic RON of 600 V GaN Switches at Variable Operating Regimes," Electronics 2023, 12, 1063]. In future work, we will conduct a more in-depth analysis of the degradation of RD,ON. This will involve varying parameters such as driving voltage, blocking voltage, temperature, and the duration/timing of the applied waveforms. These investigations will allow us to explore the degradation of device performance across different application scenarios, providing valuable insights for the physical modeling and optimization of the process.

Round 2
Reviewer 1 Report
Comments and Suggestions for Authors
1. The first page, second paragraph, lines 39-43 have been described in the first paragraph and the authors should comment around the development and challenges of p-GaN HEMTs, here are some recommended references.
[1] M. Jia et al. "High VTH and Improved Gate Reliability in P-GaN Gate HEMTs With Oxidation Interlayer," IEEE Electron Device Letters, vol. 44, no. 9, pp. 1404-1407, Sep. 2023, doi: 10.1109/LED.2023.3295064.
[2] T. Pu et al. "Normally-off AlGaN/GaN heterostructure junction field-effect transistors with blocking layers," Superlattices and Microstructures, vol. 120, pp. 448-453, 2018, doi: 10.1016/j.spmi.2018.05.063.
2. The location of different layers in the energy bands of Fig. 3 need to be labeled.
3. Give the extraction process of Dit and Ec-Et, which need to be reflected in the revised manuscript.
Comments on the Quality of English LanguageThe author should check the grammar of the manuscript carefully
Author Response
Reviewer 1 (Round 2)
- The first page, second paragraph, lines 39-43 have been described in the first paragraph and the authors should comment around the development and challenges of p-GaN HEMTs, here are some recommended references.
[12] M. Jia et al. "High VTH and Improved Gate Reliability in P-GaN Gate HEMTs With Oxidation Interlayer," IEEE Electron Device Letters, vol. 44, no. 9, pp. 1404-1407, Sep. 2023, doi: 10.1109/LED.2023.3295064.
[23] T. Pu et al. "Normally-off AlGaN/GaN heterostructure junction field-effect transistors with blocking layers," Superlattices and Microstructures, vol. 120, pp. 448-453, 2018, doi: 10.1016/j.spmi.2018.05.063.
Reply:Thank you sincerely for your valuable suggestions. We have modified the content and added references. (lines 38-52)
To address these challenges and harness the full potential of GaN HEMTs, innovative approaches were explored to manipulate the electrical properties and structural characteristics of GaN-based devices. Among these strategies were the development of non-polar a-plane channels, fluorine treatment [4], gate recesses [5], p-type GaN cap [6-10], and cascode structures. Among these approaches, the p-GaN gate structure has become the commercially available normally-off p-GaN gate HEMT due to its outstanding figure of merit and reliable normally-off functionality. For power switching applications that require normally-off characteristics, such as CMOS circuits that require safe operation and simple gate drive configurations. Therefore, there are several significant challenges facing p-GaN gate HEMTs that require improvement. Firstly, reducing gate leakage current by adding a dielectric layer under the gate is an effective strategy to reduce leakage current and increase gate drive voltage [12]. Secondly, during the growth of the p-GaN cap layer and to ensure optimal activation through MOCVD, it is necessary to incorporate an intrinsic GaN layer. This layer functions as a barrier to minimize Mg out-diffusion into the AlGaN barrier and/or GaN channel, thus preventing 2DEG degradation [13].
- The location of different layers in the energy bands of Fig. 3 need to be labeled.
Reply:Thank you sincerely for your valuable suggestions. We have marked in Figure 3(a). (lines 314)
- Give the extraction process of Dit and Ec-Et, which need to be reflected in the revised manuscript.
Reply:Thank you sincerely for your valuable suggestions. We added the extraction process of Dit and Ec-Et into the relevant equations and references. (lines 318-328)
Considering a continuum of trap levels, the conductance parameter Gp can be expressed as:
(1)
In this equation, ω represents the radial frequency and denotes the trap time constant given by the Shockley-Read-Hall statistics [32].
(2)
An approximate expression giving the Dit interface trap densities in terms of the measured maximum conductance, among A representing device area and q denoting the elementary charge. According to the conductance measurement data, Dit and its corresponding trap energy level located below the conduction band are obtained by fitting the data with equations (1) and (2) [33].
